# SDPose: Exploiting Diffusion Priors for Out-of-Domain and Robust Pose Estimation

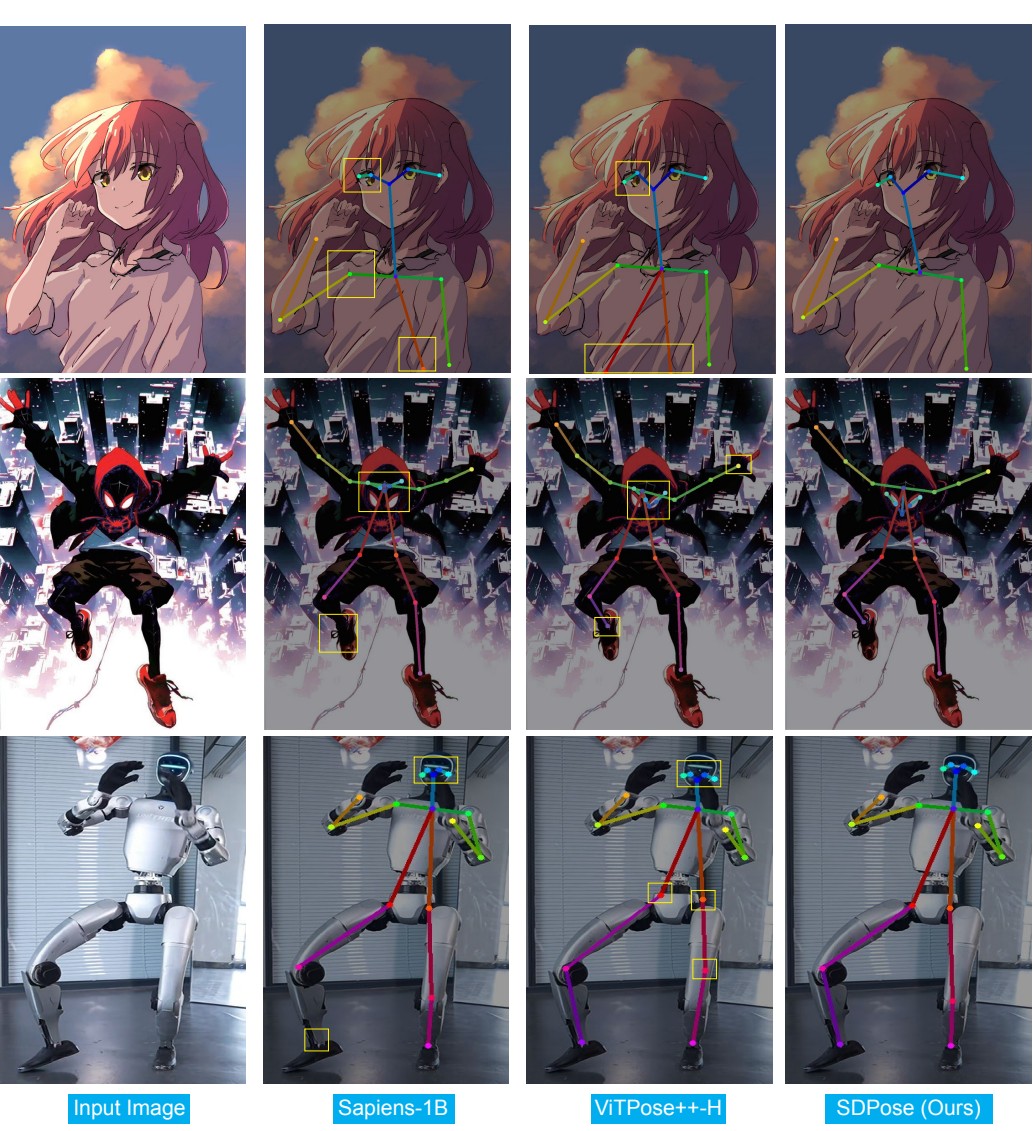

Figure 1: **SDPose: OOD-robust pose via diffusion priors.** On stylized paintings, SDPose surpasses Sapiens and ViTPose++-H, matching SoTA on COCO and setting new records on HumanArt and COCO-OOD; yellow boxes show baseline failures.

## Abstract

Pre-trained diffusion models provide rich multi-scale latent features and are emerging as powerful vision backbones. While recent works such as Marigold (Ke et al., 2024) and Lotus (He et al., 2024) adapt diffusion priors for dense prediction with strong cross-domain generalization, their poten-

tial for structured outputs (e.g., human pose estimation) remains underexplored. In this paper, we propose **SDPose**, a fine-tuning framework built upon Stable Diffusion to fully exploit pre-trained diffusion priors for human pose estimation. First, rather than modifying cross-attention modules or introducing learnable embeddings, we directly predict keypoint heatmaps in the SD U-Net's image latent space to preserve the original generative priors. Second, we map these latent features into keypoint heatmaps through a lightweight convolutional pose head, which avoids disrupting the pre-trained backbone. Finally, to prevent overfitting and enhance out-of-distribution robustness, we incorporate an auxiliary RGB reconstruction branch that preserves domain-transferable generative semantics. To evaluate robustness under domain shift, we further construct **COCO-OOD**, a style-transferred variant of COCO with preserved annotations. With just one-fifth of the training schedule used by Sapiens on COCO, SDPose attains parity with Sapiens-1B/2B on the COCO validation set and establishes a new state of the art on the cross-domain benchmarks HumanArt and COCO-OOD. Extensive ablations highlight the importance of diffusion priors, RGB reconstruction, and multi-scale SD U-Net features for cross-domain generalization, and t-SNE analyses further explain SD's domain-invariant latent structure. We also show that SDPose serves as an effective zero-shot pose annotator for controllable image and video generation.

# 1 INTRODUCTION

With the recent rise of embodied AI, video generation, and 3D asset rendering, the need for cross-domain-robust human pose estimation has become critical in robotics as well as in film, animation, and game production. Although recent advances on academic benchmarks such as MS COCO (Lin et al., 2014) using models such as DWPose (Yang et al., 2023), RTM-Pose (Jiang et al., 2023) and OpenPose (Martınez, 2019), as well as approaches leveraging large pretrained backbones such as ViTPose (Xu et al., 2022; 2023) and Sapiens (Khirodkar et al., 2024), have achieved strong in-domain accuracy, they often exhibit severe performance degradation under domain shifts and require substantial fine-tuning efforts.

Recently, pre-trained diffusion models such as Stable Diffusion (Rombach et al., 2022) have emerged as robust vision backbones. A growing body of work has shown that with fine-tuning and adaptation, diffusion priors can be repurposed for 3D generation (Cheng et al., 2023; Lin et al., 2025; Long et al., 2024), segmentation (Karmann & Urfalioglu, 2025), and dense prediction tasks (Ke et al., 2024; He et al., 2024), while consistently demonstrating strong cross-domain robustness and highlighting their potential for leveraging intra-visual multimodality in generative priors. However, their potential for structured and semantically aware outputs, particularly in human pose estimation, remains largely unexplored. Concurrent efforts like GenLoc (Wang et al., 2025a) and Diff-Tracker (Zhang et al., 2024) indicate that generative priors can benefit keypoint localization and tracking by steering learnable condition embeddings and adapting the diffusion model's cross-attention. We instead examine a complementary axis: can one rely purely on SD U-Net latent features, without attention read-outs or condition tokens, to produce reliable pose heatmaps?

To bridge this gap and investigate how SD's rich latent representations can be effectively leveraged for robust cross-domain pose estimation, our contributions are as follows:

(1) We propose **SDPose**, a fine-tuning framework with three key components: **(i) Latent-space preservation.** We operate entirely in the SD U-Net's image latent space without modifying cross-attention modules or adding learnable embeddings, thus preserving pre-trained visual semantics and feature geometry. **(ii) Lightweight pose decoder head.** We introduce a minimal decoder that maps SD U-Net features to keypoint heatmaps with only a shallow convolutional head, ensuring low overhead and minimal disturbance to the pretrained latent representations. **(iii) RGB reconstruction regularization.** We add an auxiliary reconstruction branch that regularizes fine-tuning, helping to maintain domain-transferable generative semantics and improve out-of-distribution robustness.

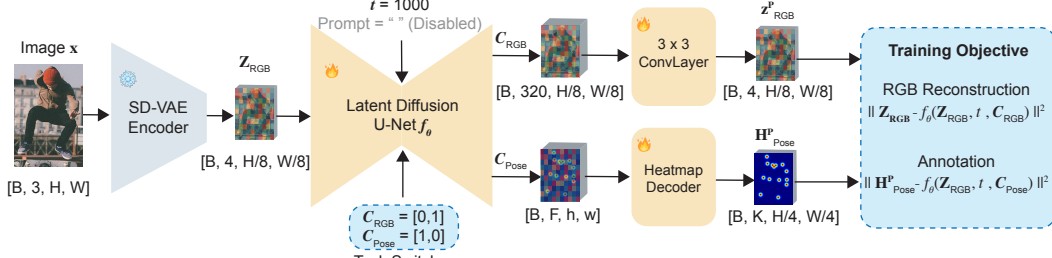

Figure 2: **Training Pipeline of SDPose.** The input RGB image is first encoded into the latent space by a pre-trained VAE. The U-Net is conditioned for multi-task learning via a class embedding. When the class label is set to [0,1], the U-Net predicts the reconstructed RGB latent; when set to [1,0], it produces features for heatmap prediction. The output layer of the U-Net is task-specific: the original convolutional output layer is retained for RGB latent reconstruction, while a lightweight heatmap decoder is used to process the U-Net's intermediate features for keypoint heatmap prediction.

(2) To systematically evaluate robustness under domain shift, we introduce **COCO-OOD**, a style-transferred extension of COCO that includes oil-painting, ukiyo-e, and color sketch domains. This dataset fills an important gap in benchmarking generalization robustness.

(3) We conduct extensive ablation studies to understand how diffusion priors and our RGB reconstruction branch contribute to cross-domain generalization in pose estimation. We further compare multi-scale features from different upsampling blocks of the SD-UNet, identifying the feature level that yields the strongest robustness under artistic domain shifts. Finally, through a latent-space comparison with Sapiens using t-SNE visualizations, we observe that SD's pretrained latent features naturally capture domain-invariant structures, which is highly beneficial for cross-domain perception tasks.

On COCO (Lin et al., 2014) and COCO-WholeBody (Jin et al., 2020), SDPose delivers in-domain performance on par with the current SoTA, Sapiens (Khirodkar et al., 2024). Under domain shift (HumanArt (Ju et al., 2023), COCO-OOD), it sets a new state of the art while using only one-fifth of Sapiens's fine-tuning epochs, highlighting the efficiency and cross-domain robustness of generative priors. Beyond quantitative benchmarks, we further demonstrate SDPose as a zero-shot pose annotator for downstream controllable generation tasks, including ControlNet-based image synthesis and video generation, where it provides reliable and qualitatively superior pose guidance.

## 2 Related Works

### 2.1 Latent Diffusion Models

Latent diffusion models (LDMs), built on DDPM and further advanced by ODE and SDE samplers (Ho et al., 2020; Song et al., 2020b;a; Lu et al., 2022; Rombach et al., 2022), have gained traction over the past few years. Classic architectures, such as the UNet-based Stable Diffusion and Diffusion Transformers (DiT) (Peebles & Xie, 2023; Esser et al., 2024), have demonstrated strong performance across diverse conditional generation tasks (Zhang et al., 2023). Pretrained on large-scale datasets such as LAION-5B (Schuhmann et al., 2022), generative models like Stable Diffusion provide rich visual priors that can be effectively leveraged for a wide range of tasks. Recent advances in flow-matching (Lipman et al., 2022; Esser et al., 2024; Xie et al., 2024) further show that latent diffusion models can achieve high-quality synthesis with only a few sampling steps. These developments highlight the power of latent generative priors as a strong visual foundation.

### 2.2 Leveraging Diffusion Priors for Prediction Tasks

A growing body of work has explored repurposing pretrained latent diffusion priors for dense prediction tasks. Marigold (Ke et al., 2024) adapts Stable Diffusion by fine-tuning only the denoising U-Net using synthetic data, delivering high-quality depth results. Later, subsequent methods such as Lotus (He et al., 2024) and GenPercept (Xu et al., 2024) both adopt a deterministic one-step fine-tuning strategy, removing

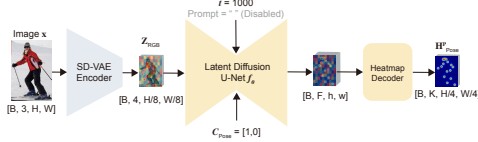

Figure 3: **SDPose Inference Pipeline.**

the multi-step stochastic diffusion process and directly predicting task annotations, which significantly improves both accuracy and inference speed. In contrast, leveraging latent diffusion priors for structured outputs (e.g., human pose) remains underexplored. Prior works such as GenLoc (Wang et al., 2025a) and Diff-Tracker (Zhang et al., 2024) freeze Stable Diffusion backbone and use learnable condition or prompt embeddings to read cross-attention maps, rather than decoding from U-Net latent features, aiming at zero-/few-shot and schema-flexible generalization. We instead remain in the image latent space, treat the SD U-Net as a multi-scale backbone, and attach a minimal convolutional head to produce keypoint heatmaps, retaining SD-native visual semantics and improves robustness under domain shift.

### 2.3 HUMAN POSE ESTIMATION

Human pose estimation is a classic and fundamental task in computer vision. Early approaches predominantly relied on CNN backbones such as HRNet (Sun et al., 2020) and CSPNeXt (Chen et al., 2024), coupled with heuristically designed decoding heads. Models like RTMPose (Jiang et al., 2023) and DWPose (Yang et al., 2023) achieved strong performance on academic benchmarks such as COCO and COCO WholeBody. However, these models exhibit limited generalization when transferring from real human figures to out-of-domain cases, such as anime characters. More recently, fine-tuned methods built on extensive pre-trained vision backbones, such as ViTPose (Xu et al., 2022; 2023) and Sapiens (Khirodkar et al., 2024), have achieved SoTA results on standard benchmarks, demonstrating the benefit of leveraging pre-trained foundation models for pose estimation. Nevertheless, these methods incur high fine-tuning costs, as they require large task-specific datasets and lengthy training schedules to achieve competitive performance. In this paper, we demonstrate that fine-tuning the Stable Diffusion pipeline with minimal architectural modifications can address both the generalization gap and the high fine-tuning cost.

### 3 PRELIMINARIES

#### 3.1 HEATMAP REPRESENTATION AND UNBIASED DATA PROCESSING (UDP)

Let $(x_i, y_i)$ denote the $i$-th ground-truth keypoint in an $H \times W$ image. The standard heatmap representation encodes each keypoint as

$$H_i(u, v) = \exp\left(-\frac{(u - x_i)^2 + (v - y_i)^2}{2\sigma^2}\right), \quad (\hat{x}_i, \hat{y}_i) = \arg\max_{u,v} H_i(u, v).$$

While widely adopted, this discrete pixel-space formulation suffers from quantization bias: predicted coordinates become misaligned under flips, scales, or rotations since the argmax operation only yields integer positions. To address this issue, we adopt the Unbiased Data Processing (UDP) method (Huang et al., 2020), which removes quantization bias by estimating keypoints in a continuous domain. Following common practice, the heatmap is generated at one-quarter resolution of the input image, which balances localization accuracy with computational efficiency.

#### 3.2 PARAMETERIZATION FOR LATENT DIFFUSION MODEL

Traditional latent diffusion models (LDMs) (Ho et al., 2020) adopt the $\epsilon$-prediction parameterization, where the denoiser $f_\theta$ is trained to predict the Gaussian noise $\epsilon_t$ added at

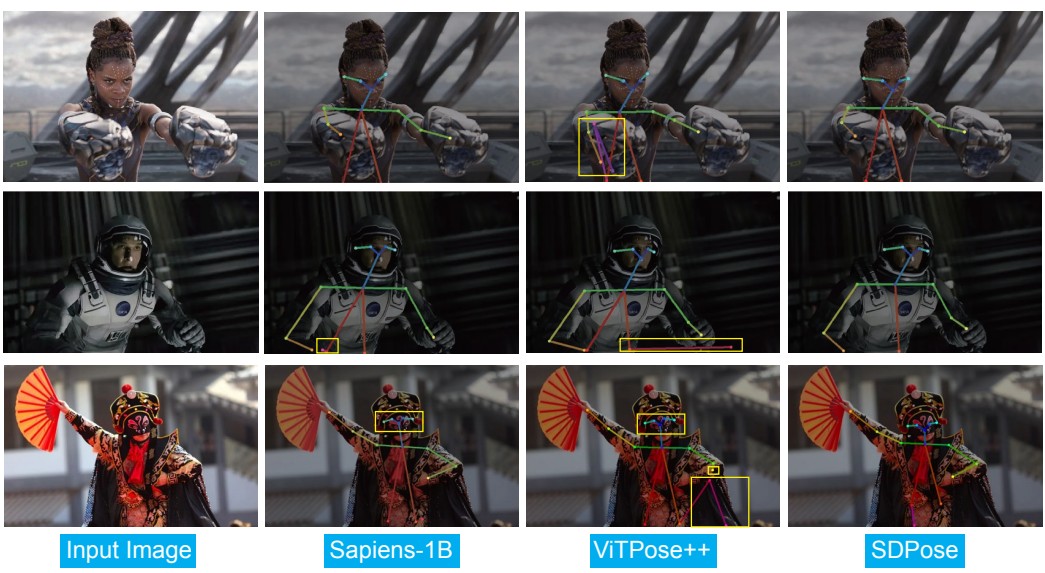

Figure 4: **Qualitative results on real-world photographs.** The yellow boxes highlight regions where baselines fail to predict accurate poses.

timestep $t$:

$$\hat{\epsilon}_t = f_\theta(z_t, t),$$

with $z_t$ denoting the noisy latent at step $t$. The clean latent $x_0$ can then be recovered by

$$\hat{x}_0 = \frac{z_t - \sqrt{1 - \alpha_t}\,\hat{\epsilon}_t}{\sqrt{\alpha_t}},$$

where $\alpha_t = \prod_{s=1}^{t}(1 - \beta_s)$ is the cumulative product of the noise schedule.

However, Lotus (He et al., 2024) shows that $\epsilon$-prediction injects unnecessary stochastic variation, which accumulates across multiple denoising steps, degrading dense prediction quality. Lotus therefore advocates a **deterministic** adaptation, directly predicting the clean annotation latent $x_0$ in a single step:

$$\hat{x}_0 = g_\theta(z_T),$$

where $T$ is a fixed timestep and $g_\theta$ is fine-tuned U-Net applied once. This formulation eliminates the prediction variance introduced by multiple-step denoising, simplifies optimization, and significantly accelerates inference. In our approach, we similarly avoid the diffusion chain and adopt this $x_0$-prediction design.

## 4 METHODOLOGY

### 4.1 LEVERAGING THE MULTI-SCALE LATENT FEATURES FOR POSE ESTIMATION

We directly leverage the multi-scale latent features of SD U-Net for the pose estimation task. The input image is encoded by the frozen SD-VAE encoder and then fed into the SD U-Net, from which we extract multi-scale features at the upsampling stage. These multi-scale features provide rich and robust visual representations for downstream keypoint prediction. The specific feature level used for each task configuration is discussed in Sec. 5.5.2.

### 4.2 THE U-NET CONVOLUTIONAL OUTPUT LAYER FORMS AN INFORMATION BOTTLENECK

Stable Diffusion's U-Net outputs a 4-channel latent $z \in \mathbb{R}^{4 \times h \times w}$ for the VAE through a single convolutional layer. In contrast, pose estimation requires $K$-channel heatmaps

Table 1: **Quantitative comparison across COCO, HumanArt, COCO-OOD Monet, and COCO-OOD Ukiyo-e.** All models are trained on COCO. Full quantitative comparison across various models are in the supplementary.

| Model Variant | Model | Pre-trained Backbone | Params | COCO | | HumanArt | | COCO-OOD Monet | | COCO-OOD Ukiyo-e | |
|---|---|---|---|---|---|---|---|---|---|---|---|
| | | | | AP | AR | AP | AR | AP | AR | AP | AR |
| **Body** | Sapiens-1B (Khirodkar et al., 2024) | Sapiens ViT | 1.169B | 82.1 | 85.9 | 64.3 | 67.4 | 58.8 | 63.3 | 61.5 | 66.2 |
| | Sapiens-2B (Khirodkar et al., 2024) | Sapiens ViT | 2.163B | **82.2** | **86.0** | 69.6 | 72.2 | 59.6 | 64.0 | 62.3 | 66.8 |
| | GenLoc (Wang et al., 2025a) | Stable Diffusion-v1.5 | 0.95B | 77.6 | 80.7 | 67.0 | 70.8 | N/A | N/A | N/A | N/A |
| | **SDPose (Ours)** | Stable Diffusion-v2 | 0.95B | 81.3 | 85.2 | **71.2** | **73.9** | **64.3** | **68.9** | **66.0** | **70.7** |
| **Wholebody** | Sapiens-1B (Khirodkar et al., 2024) | Sapiens ViT | 1.169B | 72.7 | 79.2 | N/A | N/A | 38.7 | 46.8 | 40.5 | 49.4 |
| | Sapiens-2B (Khirodkar et al., 2024) | Sapiens ViT | 2.163B | **74.4** | **81.0** | N/A | N/A | 44.4 | 53.0 | 46.6 | 55.8 |
| | **SDPose (Ours)** | Stable Diffusion-v2 | 0.95B | 71.5 | 78.4 | N/A | N/A | **46.6** | **54.8** | **47.7** | **56.4** |

$H \in \mathbb{R}^{K \times H' \times W'}$ with $K \gg 4$, making the 4-channel latent a severe information bottleneck. To address this, we replace the original 4-channel head with a lightweight heatmap decoder (Xiao et al., 2018). The decoder consists of deconvolution layers for upsampling, followed by convolutions that output $K$-channel heatmaps (Fig. 2). This modification removes the bottleneck and shortens the supervision path to keypoints.

### 4.3 Auxiliary RGB Reconstruction

To preserve the fine-detail representation capability of diffusion priors and to avoid overfitting to the pose estimation domain, we adopt the *Detail Preserver* strategy from Lotus (He et al., 2024). Concretely, we introduce a class embedding $C \in \{C_{\mathrm{RGB}}, C_{\mathrm{Pose}}\}$ that controls the behavior of the denoising U-Net $f_\theta$. When $C_{\mathrm{RGB}}$ is provided, the network is trained to reconstruct the RGB latent $z_{\mathrm{RGB}}$; when $C_{\mathrm{Pose}}$ is provided, it learns to reconstruct the ground-truth heatmap $H_{\mathrm{Pose}}$. The overall objective is

$$L = \|z_{\mathrm{RGB}} - f_\theta(z_{\mathrm{input}}, t, C_{\mathrm{RGB}})\|^2 + \|H_{\mathrm{Pose}} - f_\theta(z_{\mathrm{input}}, t, C_{\mathrm{Pose}})\|^2,$$

where $z_{\mathrm{input}}$ is the latent encoded from the input image by the SD-VAE, and $t$ is fixed to $t = 1000$ in our experiments.

### 4.4 Inference

As illustrated in Fig. 3, the input RGB image $x$ is encoded by the SD-VAE into the latent representation $z_{\mathrm{RGB}}$. The latent diffusion U-Net then performs a single-step regression with the timestep fixed at $t = 1000$, using the class label $C_{\mathrm{Pose}}$ to execute the pose estimation task. The text condition is disabled by feeding an empty text embedding to the U-Net.

## 5 Experiments

### 5.1 Experiment Settings

**Implementation Details.** We train SDPose based on Stable Diffusion V2 (Rombach et al., 2022), with text conditioning disabled. During training, we fix the timestep $t = 1000$. For more details, please see the supplementary materials.

**Training Datasets.** We train two variants, SDPose Body (17-keypoints) and SDPose Wholebody (133-keypoints), on MS COCO (Lin et al., 2014) and COCO-WholeBody (Jin et al., 2020), respectively. All images are processed using standard top-down augmentations, with the input resolution set to $1024 \times 768$. Further details are provided in the supplementary materials.

**Validation Datasets and Metrics.** (1) For the Body variant, we evaluate SDPose on MS COCO (Lin et al., 2014) for real-world images, and on HumanArt (Ju et al., 2023) and COCO-OOD for cross-domain benchmarks. (2) For the Wholebody variant, we evaluate SDPose on COCO-WholeBody (Jin et al., 2020) and the extended COCO-OOD. Further details of the evaluation datasets and metrics are provided in the supplementary materials.

**COCO-OOD.** To complement HumanArt and enable OOD evaluation with matched content and labels, we translate all COCO val images into three artistic domains: Monet-style

paintings using the official StyTR2 framework (Deng et al., 2022), ukiyo-e style using the official CycleGAN implementation (Zhu et al., 2017), and color-sketch style using Nano Banana (Nano Banana). For all stylized images, we reuse the original COCO val annotations (bounding boxes and keypoints). Please refer to the supplementary materials for details.

## 5.2 Quantitative and Qualitative Comparison on Real-World Scenes

For the Body variant, SDPose achieves 81.3 AP / 85.2 AR on the COCO validation set (Table 1) with only 40 training epochs using a 0.95B SD-v2 backbone. It matches the accuracy of Sapiens (82.1–82.2 AP) despite requiring 5× fewer epochs and a smaller backbone, and surpasses GenLoc (+3.7 AP, +4.5 AR). Figure 4 further illustrates robustness on real-world photos, where SDPose rivals Sapiens and corrects its failure cases (e.g., Sichuan opera eye keypoints). For the Wholebody variant, SDPose achieves competitive performance with Sapiens-1B on

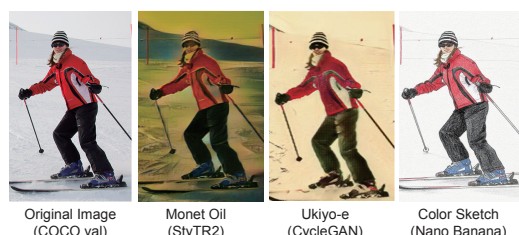

Figure 5: **Illustration of the COCO-OOD dataset.**

the COCO-WholeBody validation set. Further details are provided in the supplementary materials.

## 5.3 SDPose's Strong OOD Robustness

In this section, we demonstrate the superior OOD robustness of SDPose using quantitative evaluation on HumanArt and our COCO-OOD benchmark. As shown in Table 1, SDPose achieves state-of-the-art results on HumanArt and COCO-OOD with fewer training epochs and a smaller parameter budget. On COCO-OOD WholeBody, SDPose continues to demonstrate strong out-of-domain robustness. As shown in Fig. 1, SDPose achieves more accurate body pose estimation across diverse animation styles and humanoid robots compared with baseline models. Additional qualitative results on whole-body pose estimation in stylized paintings are provided in the supplementary materials.

## 5.4 Why Diffusion Priors Exhibit Strong OOD Robustness?

In this subsection, we investigate why the diffusion-based prior (Stable Diffusion v2) (Rombach et al., 2022) exhibits stronger cross-domain generalization than the human-centric Sapiens ViT backbone (Khirodkar et al., 2024). We compare latent feature distributions from the pretrained Sapiens-1B ViT and Stable Diffusion v2 U-Net, as well as their pose-finetuned counterparts, Sapiens-1B Pose and our SDPose model (finetuned on the last four upsampling blocks). We sample 300 person instances from the COCO and COCO-OOD validation sets. For each instance, we crop the person region using the ground-truth bounding box and collect four stylistic variants: the original COCO image plus its ukiyo-e, Monet-oil, and color-sketch versions from COCO-OOD, yielding 1200 crops in total. These images are passed through the corresponding backbones to extract latent features, on which we run t-SNE (Maaten & Hinton, 2008) and compute silhouette scores (Rousseeuw, 1987).

Fig. 6(a–e) show the t-SNE visualizations of the *pretrained* priors, and Fig. 6(f–j) show the *pose-finetuned* models. In Fig. 6(a), the pretrained Sapiens features form clear style-driven clusters (silhouette by style = 0.3469) with a negative silhouette by person-instance ($-0.1608$), indicating that its representation is dominated by artistic appearance rather than instance structure. In contrast, Stable Diffusion U-Net features in Fig. 6(b–e) exhibit much weaker style separation and gradually stronger person-instance coherence; deeper upsampling blocks yield increasingly person-consistent and style-invariant distributions, reflected by rising person-instance silhouettes (up to $\approx 0.22$) and near-zero style silhouettes. After pose finetuning, both Sapiens-Pose (Fig. 6(f)) and SDPose (Fig. 6(g–j)) produce visible person-instance clusters, but SDPose features are noticeably tighter and more focused, with the mid-level SDPose blocks achieving the highest person silhouettes ($\approx 0.45$–0.48) while

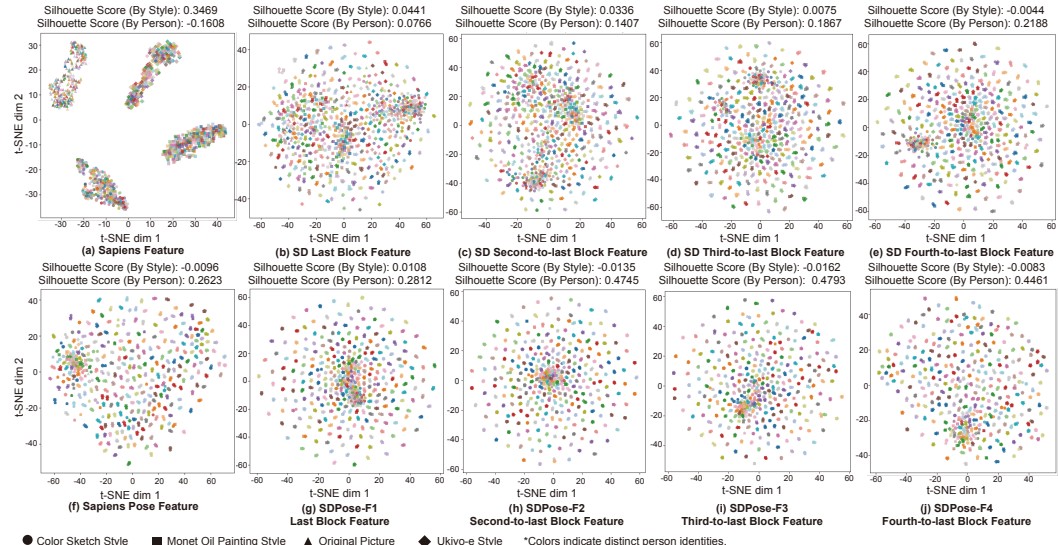

Figure 6: **t-SNE visualization of features from Sapiens ViT, Stable Diffusion U-Net blocks, Sapiens Pose, and SDPose across four visual domains.** Each point corresponds to an image sample; colors represent person instances and marker shapes denote artistic styles.

keeping style silhouettes close to zero. Overall, the SD U-Net (especially after SDPose fine-tuning) provides more instance-coherent and style-disentangled representations than Sapiens, which explains why SD-based features offer stronger cross-domain robustness under artistic style shifts.

## 5.5 Ablation Study

### 5.5.1 Ablation on Auxiliary Reconstruction Task and Diffusion Priors

We conduct ablations to validate our designs. For the "w/o diffusion priors" variant, we train the U-Net from scratch (no pretrained priors). For the "w/o RGB recon." variant, we disable only the auxiliary RGB reconstruction branch; all other settings remain identical. From Table 2, two trends emerge. First, removing the RGB branch yields a consistent but modest AP/AR drop on COCO that becomes more pronounced on HumanArt and COCO-OOD, indicating that the auxiliary reconstruction acts as a useful regularizer and improves robustness under domain shift. Second, removing diffusion priors causes a much larger degradation, especially on the OOD benchmarks, highlighting that the pretrained generative priors are the primary source of SDPose's generalization.

### 5.5.2 Ablation on Multi-scale Features from the SD U-Net

Prior work (Liu et al., 2023b; 2024; 2023a; Wang et al., 2025b) suggests that penultimate features often transfer better than final ones. As shown in Table 2, we evaluate SD U-Net upsampling features from the last four blocks. On COCO (17-keypoint), the last-block (F1) and penultimate (F2) features perform very similarly, and F1 is slightly better on both COCO and HumanArt, which contain a mixture of natural and artistic content. In contrast, on COCO-OOD Monet, where the domain shift is purely stylistic, F2 achieves the best performance, suggesting that it captures style-invariant cues more effectively than F1. For the 133-keypoint WholeBody setting, F2 consistently outperforms all other feature levels on both COCO-WholeBody and COCO-OOD, indicating that it offers a better balance between semantic robustness and spatial detail in this more fine-grained regime. Together with the latent-space analysis in Section 5.4, these results show that deeper SD features (F3 and F4) encode even stronger instance-consistent, style-invariant semantics but operate at much

Table 2: **Ablation studies on diffusion priors, RGB reconstruction, and U-Net feature selection.** All experiments are trained on COCO with 40 epochs (body-17 keypoints) or 42 epochs (wholebody-133 keypoints).

| Model Variant | Ablation Setting | COCO | | HumanArt | | COCO-OOD Monet | |
|---|---|---|---|---|---|---|---|
| | | AP | AR | AP | AR | AP | AR |
| SDPose-Body | Full Model (Last Block) | **81.3** | **85.2** | **71.2** | **73.9** | 63.5 | 68.2 |
| | w/o RGB Reconstruction | 80.8 (-0.5) | 84.9 (-0.3) | 69.8 (-1.4) | 72.6 (-1.3) | 62.5 (-1.0) | 67.3 (-0.9) |
| | w/o Diffusion Priors | 74.9 (-6.4) | 79.4 (-5.8) | 53.8 (-17.4) | 58.0 (-15.9) | 52.7 (-10.8) | 57.9 (-10.3) |
| | Last Block (F1) | **81.3** | **85.2** | **71.2** | **73.9** | 63.5 | 68.2 |
| | Second-to-last Block (F2) | 81.1 | 85.0 | 70.4 | 73.3 | **64.3** | **68.9** |
| | Third-to-last Block (F3) | 81.0 | 85.1 | 70.6 | 73.3 | 63.5 | 68.2 |
| | Fourth-to-last Block (F4) | 79.2 | 83.4 | 65.0 | 68.1 | 58.1 | 62.8 |
| SDPose-Wholebody | Last Block (F1) | 70.5 | 77.5 | N/A | N/A | 44.7 | 53.0 |
| | Second-to-last Block (F2) | **71.5** | **78.4** | N/A | N/A | **46.6** | **54.8** |
| | Third-to-last Block (F3) | 70.4 | 77.6 | N/A | N/A | 45.5 | 53.8 |
| | Fourth-to-last Block (F4) | 64.6 | 72.1 | N/A | N/A | 37.1 | 45.4 |

coarser spatial resolutions (H/16 and H/32), which removes the geometric details needed for accurate keypoint localization. In practice, we therefore use F1 as the default feature level for the SDPose Body model (favoring slightly better in-domain and mixed-domain performance) and F2 as the default feature level for the SDPose WholeBody model (favoring consistently stronger performance under both in-domain and out-of-domain settings).

## 6 DOWNSTREAM APPLICATIONS

### 6.1 BETTER POSE-GUIDED IMAGE GENERATION

For human or humanoid character generation, an accurate skeleton is essential for transferring poses between characters. Traditional pose estimators often fail to precisely capture the skeletons of art-based human or humanoid characters. Our method provides a generalizable pose estimation approach that can benefit animation production. As shown in Fig. 7, we compare ControlNet (Zhang et al., 2023) outputs using DWPose as the baseline pose annotator. Notably, our SDPose yields more precise and detailed skeletons than DWPose, enabling reliable pose transfer and high-quality image generation for artistic characters.

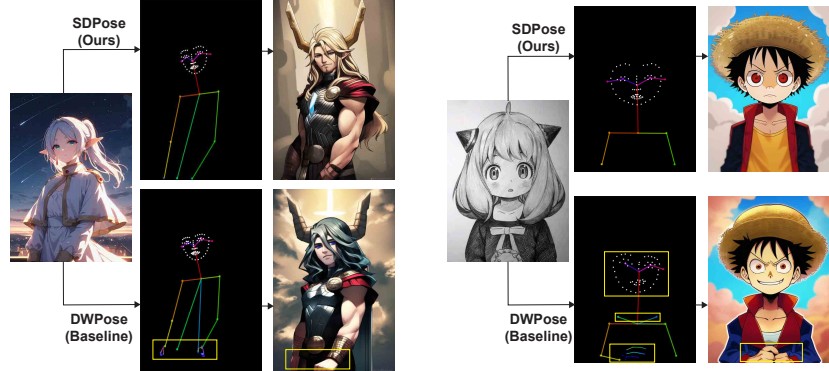

Figure 7: **Visualization of pose-guided image generation results.** The lower images illustrate results from the baseline, which combines a pre-trained ControlNet with the DWPose estimator. In comparison, the upper images show results obtained using our SDPose as the pose annotator. Yellow boxes highlight baseline failures. Prompts, random seeds, and other settings are kept identical for fairness.

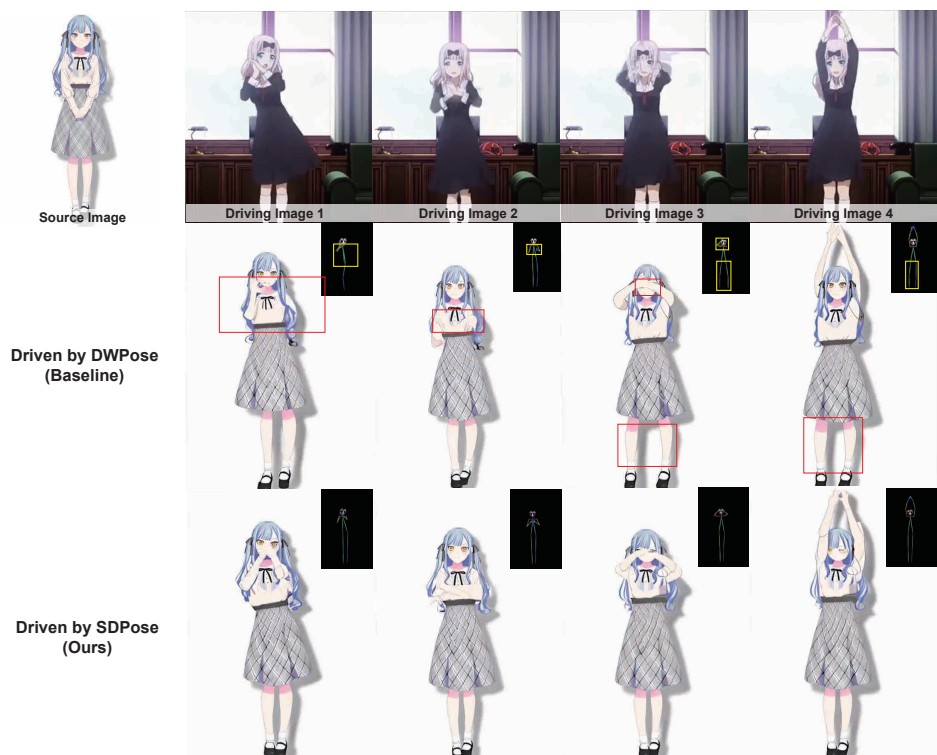

Figure 8: **Qualitative comparison for pose-controlled video generation in the wild.** The first row shows the source image and frames from the driving video. The second row shows output video frames generated from the pose sequence estimated by the baseline model DWPose, while the third row shows the results guided by our SDPose. Red boxes highlight failures in the generated video, and yellow boxes highlight errors in pose estimation.

## 6.2 POSE-GUIDED VIDEO GENERATION

Recent advances in controlled video generation have gained significant traction (Hu, 2024; Guo et al., 2023; Kim et al., 2024). Despite the progress of video generation models in producing higher-quality outputs, extracting reliable control conditions remains critical for achieving high-quality results. As shown in Fig. 8, our SDPose provides more accurate poses for the driving frames, enabling more reliable pose-sequence transfer from animations to animations. Video frames are generated by Moore-Animated Anyone[1].

## 7 CONCLUSION

In this paper, we present **SDPose**, an SD-native fine-tuning framework for human pose estimation. SDPose preserves the original U-Net with only lightweight task-specific components and adapts diffusion latent priors for keypoint prediction through an RGB reconstruction branch and a heatmap decoder. We further introduce **COCO-OOD**, a style-transferred extension of COCO for evaluating robustness under domain shifts. With only one-fifth of the fine-tuning cost of Sapiens and a smaller backbone, SDPose matches its in-domain accuracy on COCO and achieves state-of-the-art results on COCO-OOD and HumanArt. Our ablations and latent-space analyses show that diffusion priors and multi-scale SD features naturally encode domain-invariant structure, enabling strong generalization in both pose estimation and zero-shot pose annotation for controllable generation.

---

[1]https://github.com/MooreThreads/Moore-AnimateAnyone

Ethics Statement

This work builds upon publicly available datasets (COCO, COCO Wholebody), all of which have established licenses and annotation protocols. No private or personally identifiable information is used. Our method focuses on improving the robustness of pose estimation under domain shifts, which can benefit applications such as animation and embodied AI.

Reproducibility Statement

We have made every effort to ensure reproducibility. All datasets used are publicly available, and we detail dataset splits, preprocessing steps, and evaluation protocols in Sec. 4. Our training settings, hyperparameters, and model architectures are fully described in Sec. 4, Sec. 5 and Appendix A. Code and scripts to reproduce our experiments will be released upon publication.

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

SUPPLEMENTARY MATERIALS FOR **SDPose**: EXPLOITING DIFFUSION PRIORS FOR OUT-OF-DOMAIN AND ROBUST POSE ESTIMATION

## A EXPERIMENT SETTINGS

### A.1 IMPLEMENTATION DETAILS

We train SDPose on the COCO-2017 person keypoints *train2017* split only (no extra data), with text prompts disabled. The diffusion timestep is fixed at $t = 1000$. We use AdamW with a learning rate of $3 \times 10^{-5}$. All experiments are run on 8 NVIDIA A100-NVLink GPUs with a total batch size of 128, without gradient accumulation. Inputs are resized to $1024 \times 768$ with standard top-down augmentations. The 17-keypoint model is trained for 40 epochs (approximately 3 days), and the 133-keypoint model for 42 epochs (approximately 3 days and a half).

### A.2 TRAINING DATASETS

**COCO 2017 Keypoint Detection** We train the 17-keypoint variant on the COCO-2017 person keypoint detection dataset (Lin et al., 2014). The full COCO release contains more than 200,000 images and about 250,000 person instances. Person keypoint annotations follow the 17-point format (nose, eyes, ears, shoulders, elbows, wrists, hips, knees, and ankles).

**COCO Wholebody** To further evaluate large-scale whole-body keypoint estimation, we adopt COCO-WholeBody (Jin et al., 2020), an extended benchmark built on top of COCO images. COCO-WholeBody augments the original 17 body joints with fine-grained annotations of foot (6 keypoints), face (68 keypoints), and hands (42 keypoints for hands), resulting in a total of 133 keypoints per person. The dataset provides consistent whole-body annotations across the same training and validation splits as COCO-2017, enabling both fair comparison with standard pose estimation methods and comprehensive evaluation under the whole-body setting.

### A.3 AUGMENTATION DETAILS

The training pipeline first loads the input image and computes the bounding box center and scale. It applies random horizontal flipping, half-body augmentation, and random bounding box transformations. The image is then affine-transformed to the target input resolution using UDP (Huang et al., 2020). Albumentations-based augmentations are then applied, including Gaussian blur ($p = 0.1$), median blur ($p = 0.1$), and coarse dropout ($p = 1.0$, with up to one hole of size 20%–40% of the image).

### A.4 EVALUATION DATASETS AND METRICS

**Evaluation Datasets**

**COCO 2017 Keypoint Detection.** For in-domain evaluation, we use the COCO-2017 validation set (Lin et al., 2014) annotated with 17 body keypoints, bounding boxes, and visibility flags. Following the standard top-down evaluation protocol, we generate person crops from the COCO-released detection results (`COCO_val2017_detections_AP_H_70_person.json`), and report COCO keypoint AP/AR on this diverse in-the-wild dataset.

**HumanArt.** We use HumanArt (Ju et al., 2023) as an cross domain benchmark: 50k human-centric images across 20 scenarios (5 natural, 15 artistic—oil painting, sculpture, cartoon, sketch, stained glass, Ukiyo-e, watercolor, etc.) with annotations for boxes and 2D keypoints. We follow the official protocol and report keypoint AP/AR to assess robustness under artistic domain shift.

**COCO-WholeBody (133-keypoint whole-body).** We train and evaluate a 133-keypoint variant on COCO-WholeBody (Jin et al., 2020), which shares COCO's

train/val split. Each person has 133 keypoints (17 body, 6 foot, 68 face, 42 hand) plus boxes for person/face/left/right hand. We follow the official protocol and report Whole-Body AP and part-wise AP (body/foot/face/hand). This dataset spans diverse in-the-wild scenes and stresses fine-grained articulation, complementing COCO for structured keypoint evaluation.

**Metrics** We follow the standard COCO keypoint evaluation protocol, which is based on the Object Keypoint Similarity (OKS). For each keypoint $i$, the similarity is defined as

$$KS_i = \exp\left(-\frac{d_i^2}{2s^2 k_i^2}\right),$$

where $d_i$ denotes the Euclidean distance between predicted and ground-truth keypoints, $s$ is the object scale (square root of the segmentation area), and $k_i$ is a per-keypoint constant controlling falloff. The OKS for an instance is the average $KS_i$ over visible keypoints:

$$OKS = \frac{\sum_i KS_i \cdot \delta(v_i > 0)}{\sum_i \delta(v_i > 0)},$$

where $v_i$ is the visibility flag. Using OKS as the matching criterion, COCO computes Average Precision (AP) as the mean precision over OKS thresholds $[0.50 : 0.05 : 0.95]$, and Average Recall (AR) analogously as the mean recall across the same thresholds.

## A.5 DETAILS OF COCO-OOD

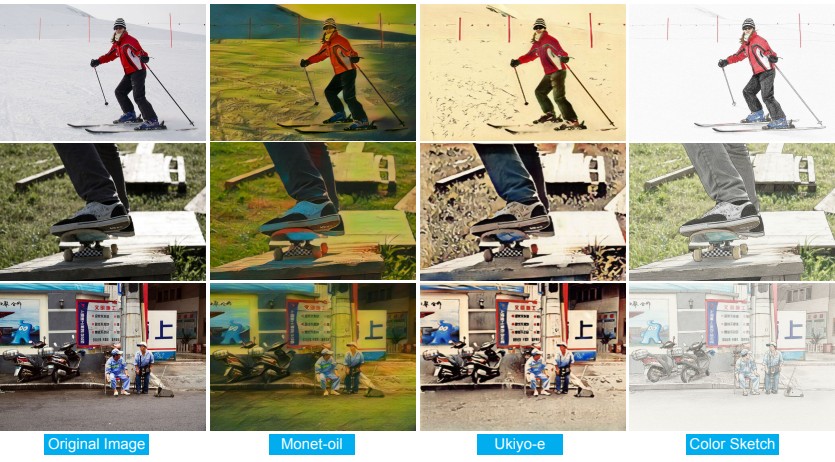

Original Image    Monet-oil    Ukiyo-e    Color Sketch

Figure 9: **COCO-OOD visualizations.** We create an OOD split of the COCO validation set by stylizing images with CycleGAN, StyTr2, and Nano-Banana into a Monet-like oil-painting domain, enabling robustness evaluation under appearance shifts.

To complement HumanArt and enable out-of-distribution evaluation under matched content and annotations, we construct COCO-OOD (Fig. 9) by applying artistic style transfer to the original COCO images while preserving human bounding boxes and keypoints. We generate three stylistic variants, each capturing different types of artistic domain shifts. Importantly, for fair comparison and to avoid introducing priors from large-scale pretrained diffusion models, we intentionally adopt the earlier CycleGAN (Zhu et al., 2017) and StyTr2 (Deng et al., 2022) framework rather than more recent style transfer approaches.

For the Monet oil-painting variant, we employ the StyTr2 framework, which produces high-fidelity oil-painting textures and color palettes while maintaining the overall scene geometry. For the Ukiyo-e variant, we adopt the official CycleGAN implementation to translate natural photographs into the Ukiyo-e domain. Finally, we also explore a Color Sketch variant generated using the Nano-Banana model (Nano Banana). While Nano-Banana preserves global shape in most cases, its stylization can occasionally introduce slight pixel-level misalignment with respect to the source images. Because such spatial deviations may compromise compatibility with COCO's keypoint annotations, we use the Color Sketch variant only as

a supplemental resource for latent-space t-SNE analysis and exclude it from quantitative evaluation. By drawing from multiple style-transfer frameworks, COCO-OOD reduces the bias introduced by any single model and produces stylistic shifts with diverse characteristics. This diversity leads to a more balanced and comprehensive benchmark. Overall, COCO-OOD maintains the geometric content of COCO while producing substantial appearance shifts across distinct artistic domains, providing a controlled and annotation-preserving testbed for assessing cross-domain robustness in pose estimation.

## B  Details of Quantitative Comparison and Additional Qualitative Comparison

### B.1  Full Quantitative Comparison on COCO

SDPose achieves 81.3 AP / 85.2 AR on the COCO validation set (Table 3) with only 40 training epochs using a 0.95B SD-v2 backbone. It matches the accuracy of Sapiens (82.1–82.2 AP) despite requiring 5× fewer epochs and a smaller backbone, and surpasses GenLoc (+3.7 AP, +4.5 AR). SDPose also outperforms ViTPose++ (+1.9 AP), which relies on multiple auxiliary datasets, while being trained solely on COCO.

Table 3: **Quantitative comparison on the COCO validation set.**

| Model | Input Size | AP | AR |
|---|---|---|---|
| SimpleBaseline (Xiao et al., 2018) | 256×192 | 73.5 | 79.0 |
| HRNet (Sun et al., 2019) | 384×288 | 76.3 | 81.2 |
| HRFormer (Yuan et al., 2021) | 256×192 | 77.2 | 82.0 |
| ViTPose-S (Xu et al., 2022) | 256×192 | 73.8 | 79.2 |
| ViTPose-B (Xu et al., 2022) | 256×192 | 75.8 | 81.1 |
| ViTPose-L (Xu et al., 2022) | 256×192 | 78.3 | 83.5 |
| ViTPose-H (Xu et al., 2022) | 256×192 | 79.1 | 84.1 |
| ViTPose++-S (Xu et al., 2023) | 256×192 | 75.8 | 81.0 |
| ViTPose++-B (Xu et al., 2023) | 256×192 | 77.0 | 82.6 |
| ViTPose++-L (Xu et al., 2023) | 256×192 | 78.6 | 84.1 |
| ViTPose++-H (Xu et al., 2023) | 256×192 | 79.4 | 84.8 |
| Sapiens-0.3B (Khirodkar et al., 2024) | 1024×768 | 79.6 | 83.6 |
| Sapiens-0.6B (Khirodkar et al., 2024) | 1024×768 | 81.2 | 84.9 |
| Sapiens-1B (Khirodkar et al., 2024) | 1024×768 | 82.1 | 85.9 |
| Sapiens-2B (Khirodkar et al., 2024) | 1024×768 | 82.2 | 86.0 |
| **SDPose (Ours)** | 1024×768 | **81.3** | **85.2** |

### B.2  Full Quantitative Comparison on HumanArt

On HumanArt (Table 4), SDPose sets a new state of the art with 71.2 AP / 73.9 AR, surpassing large-scale foundation baselines under the same COCO-only training: +1.6 AP over Sapiens-2B (69.6 AP) and +4.7 AP over ViTPose-H (66.5 AP), with consistent AR gains. Compared with traditional baselines such as RTMPose (Jiang et al., 2023) and HRNet (Sun et al., 2019), SDPose further delivers substantial improvements in AP, exceeding them by more than 14 points.

### B.3  Full Quantitative Comparison on COCO Wholebody

As shown in Table 5, SDPose achieves 71.5 AP / 78.4 AR on the COCO-WholeBody validation set. This result is highly competitive with the large-scale Sapiens (Khirodkar et al., 2024) models: while Sapiens-2B reaches 74.4 AP with over 2B parameters and long training

Table 4: **Quantitative Comparison on the HumanArt validation set.** Models trained on COCO, evaluated with GT bounding boxes. SDPose achieves new state-of-the-art performance.

| Model | AP | AP$^{50}$ | AP$^{75}$ | AR | AR$^{50}$ |
|---|---|---|---|---|---|
| RTMPose-T (Jiang et al., 2023) | 44.4 | 72.5 | 45.3 | 48.8 | 75.0 |
| RTMPose-S (Jiang et al., 2023) | 48.0 | 73.9 | 49.8 | 52.1 | 76.3 |
| RTMPose-M (Jiang et al., 2023) | 53.2 | 76.5 | 56.3 | 57.1 | 78.9 |
| RTMPose-L (Jiang et al., 2023) | 56.4 | 78.9 | 60.2 | 59.9 | 80.8 |
| ViTPose-S (Xu et al., 2022) | 50.7 | 75.8 | 53.1 | 55.1 | 78.0 |
| ViTPose-B (Xu et al., 2022) | 55.5 | 78.2 | 59.0 | 59.9 | 80.9 |
| ViTPose-L (Xu et al., 2022) | 63.7 | 83.8 | 68.9 | 67.7 | 85.9 |
| ViTPose-H (Xu et al., 2022) | 66.5 | 86.0 | 71.5 | 70.1 | 87.1 |
| HRNet-W32 (Sun et al., 2019) | 53.3 | 77.1 | 56.2 | 57.4 | 79.2 |
| HRNet-W48 (Sun et al., 2019) | 55.7 | 78.2 | 59.3 | 59.5 | 80.4 |
| Sapiens-1B (Khirodkar et al., 2024) | 64.3 | 82.1 | 67.9 | 67.4 | 83.7 |
| Sapiens-2B (Khirodkar et al., 2024) | 69.6 | 85.3 | 73.3 | 72.2 | 86.8 |
| **SDPose (Ours)** | **71.2** | **87.3** | **76.3** | **73.9** | **88.6** |

Table 5: **Quantitative comparison on the COCO-WholeBody validation set.**

(a) **Body, Feet, Face**

| Model | Body AP | Body AR | Feet AP | Feet AR | Face AP | Face AR |
|---|---|---|---|---|---|---|
| HRNet (Sun et al., 2019) | 70.1 | 77.3 | 58.6 | 69.2 | 72.7 | 78.3 |
| VitPose+-L (Xu et al., 2023) | 75.3 | - | 77.1 | - | 63.0 | - |
| VitPose+-H (Xu et al., 2023) | 75.9 | - | 77.9 | - | 63.6 | - |
| RTMPose-x (Jiang et al., 2023) | 71.4 | 78.4 | 69.2 | 81.0 | 88.8 | 92.2 |
| DWPose-l (Yang et al., 2023) | 72.2 | 78.9 | 70.4 | 81.7 | 88.7 | 92.1 |
| Sapiens-0.3B (Khirodkar et al., 2024) | 66.4 | 73.4 | 67.3 | 78.4 | 87.1 | 91.2 |
| Sapiens-0.6B (Khirodkar et al., 2024) | 74.3 | 80.2 | 79.4 | 87.0 | 89.5 | 92.9 |
| Sapiens-1B (Khirodkar et al., 2024) | 77.4 | 82.9 | 83.0 | 89.8 | 90.7 | 93.6 |
| Sapiens-2B (Khirodkar et al., 2024) | **79.2** | **84.6** | **84.1** | **90.9** | **91.2** | **93.8** |
| SDPose (Ours) | 77.9 | 83.4 | 81.5 | 88.7 | 88.5 | 92.2 |

(b) **Hands and Whole-body**

| Model | Hand AP | Hand AR | Whole AP | Whole AR |
|---|---|---|---|---|
| HRNet (Sun et al., 2019) | 51.6 | 60.4 | 58.6 | 67.4 |
| VitPose+-L (Xu et al., 2023) | 54.2 | - | 60.6 | - |
| VitPose+-H (Xu et al., 2023) | 54.7 | - | 61.2 | - |
| RTMPose-x (Jiang et al., 2023) | 59.0 | 68.5 | 65.3 | 73.3 |
| DWPose-l (Yang et al., 2023) | 62.1 | 71.0 | 66.5 | 74.3 |
| Sapiens-0.3B (Khirodkar et al., 2024) | 58.1 | 67.1 | 62.0 | 69.4 |
| Sapiens-0.6B (Khirodkar et al., 2024) | 65.4 | 74.0 | 69.5 | 76.3 |
| Sapiens-1B (Khirodkar et al., 2024) | 69.2 | 77.1 | 72.7 | 79.2 |
| Sapiens-2B (Khirodkar et al., 2024) | **70.4** | **78.1** | **74.4** | **81.0** |
| SDPose (Ours) | 65.2 | 74.0 | 71.5 | 78.4 |

schedules, SDPose attains comparable accuracy with a smaller 0.95B backbone trained for only 42 epochs. In terms of sub-part analysis, our method closely matches Sapiens-2B on body (77.9 vs. 79.2) and feet (81.5 vs. 84.1), while maintaining strong performance on face (88.5 vs. 91.2). The largest gap appears on hand keypoints (65.2 vs. 70.4), reflecting the intrinsic difficulty of fine-grained articulation. Nevertheless, compared with classical baselines such as HRNet (Sun et al., 2019), RTMPose (Jiang et al., 2023), or DWPose (Yang et al., 2023), SDPose shows substantial gains of +6–12 AP across whole-body evaluation.

Table 6: **Quantitative Comparison on the COCO-OOD-Monet Wholebody validation set.**

(a) **Body, Feet, Face**

| Model | Body AP | Body AR | Feet AP | Feet AR | Face AP | Face AR |
|---|---|---|---|---|---|---|
| Sapiens-1B | 52.1 | 58.6 | 55.9 | 66.2 | 57.6 | 63.0 |
| Sapiens-2B | 59.9 | 65.8 | 63.8 | 72.4 | 58.4 | 64.2 |
| SDPose (Ours) | **60.0** | **66.1** | **62.5** | **72.0** | **64.2** | **69.9** |

(b) **Hands and Whole-body**

| Model | L-Hand AP | L-Hand AR | R-Hand AP | R-Hand AR | Whole AP | Whole AR |
|---|---|---|---|---|---|---|
| Sapiens-1B | 43.1 | 52.0 | 41.5 | 50.6 | 38.7 | 46.8 |
| Sapiens-2B | 48.2 | 56.8 | 46.8 | 55.3 | 44.4 | 53.0 |
| SDPose (Ours) | 46.3 | 55.2 | 44.9 | 54.4 | **46.6** | **54.8** |

### B.4 FULL QUANTITATIVE COMPARISON ON COCO-OOD WHOLEBODY

Detailed whole-body pose estimation results on COCO-OOD Monet are reported in Table 6. Breaking down by body part, SDPose matches or surpasses Sapiens on the most stable regions: body (60.0 AP vs. 59.9) and feet (62.5 vs. 63.8), and delivers a notable margin on face landmarks (+5.8 AP, 64.2 vs. 58.4), highlighting the reliability of SD-native features under appearance shifts. For hands, which are the most challenging due to fine-grained articulation and limited resolution, SDPose attains 46.3/44.9 AP on left/right hands, respectively, remaining competitive but still slightly behind Sapiens-2B (48.2/46.8). Nevertheless, the overall whole-body AP/AR of 46.6/54.8 establishes SDPose as the most robust framework under OOD whole-body evaluation. Detailed whole-body pose estimation results on COCO-OOD Ukiyo-e are presented in Table 7. Breaking down by region, SDPose remains competitive with Sapiens-2B on the more structurally stable parts: body (61.2 AP vs. 62.2) and feet (64.7 vs. 67.3), and shows a clear advantage on face landmarks (+4.8 AP, 64.7 vs. 59.9), reflecting the stronger style-invariant geometry encoded by diffusion U-Net features under artistic shifts. For hands—the most challenging subset due to fine-scale pose variation—SDPose achieves 45.9/44.1 AP on left/right hands, remaining close to Sapiens-2B (49.8/46.7) despite the latter's larger model capacity. Importantly, SDPose attains the highest whole-body AP/AR (47.7/56.4), demonstrating consistently stronger robustness than both Sapiens-1B and Sapiens-2B when evaluated under Ukiyo-e style transformations.

Table 7: **Quantitative Comparison on the COCO-OOD-Ukiyoe Wholebody validation set.**

(a) **Body, Feet, Face**

| Model | Body AP | Body AR | Feet AP | Feet AR | Face AP | Face AR |
|---|---|---|---|---|---|---|
| Sapiens-1B | 54.0 | 60.9 | 57.9 | 69.9 | 58.7 | 65.3 |
| Sapiens-2B | **62.2** | **68.2** | **67.3** | **76.5** | 59.9 | 66.9 |
| SDPose (Ours) | 61.2 | 67.5 | 64.7 | 75.0 | **64.7** | **71.2** |

(b) **Hands and Whole-body**

| Model | L-Hand AP | L-Hand AR | R-Hand AP | R-Hand AR | Whole AP | Whole AR |
|---|---|---|---|---|---|---|
| Sapiens-1B | 43.7 | 52.8 | 40.9 | 50.6 | 40.5 | 49.4 |
| Sapiens-2B | **49.8** | **58.7** | **46.7** | **55.7** | 46.6 | 55.8 |
| SDPose (Ours) | 45.9 | 55.7 | 44.1 | 54.3 | **47.7** | **56.4** |

## B.5 QUALITATIVE COMPARISON FOR WHOLE-BODY POSE ESTIMATION

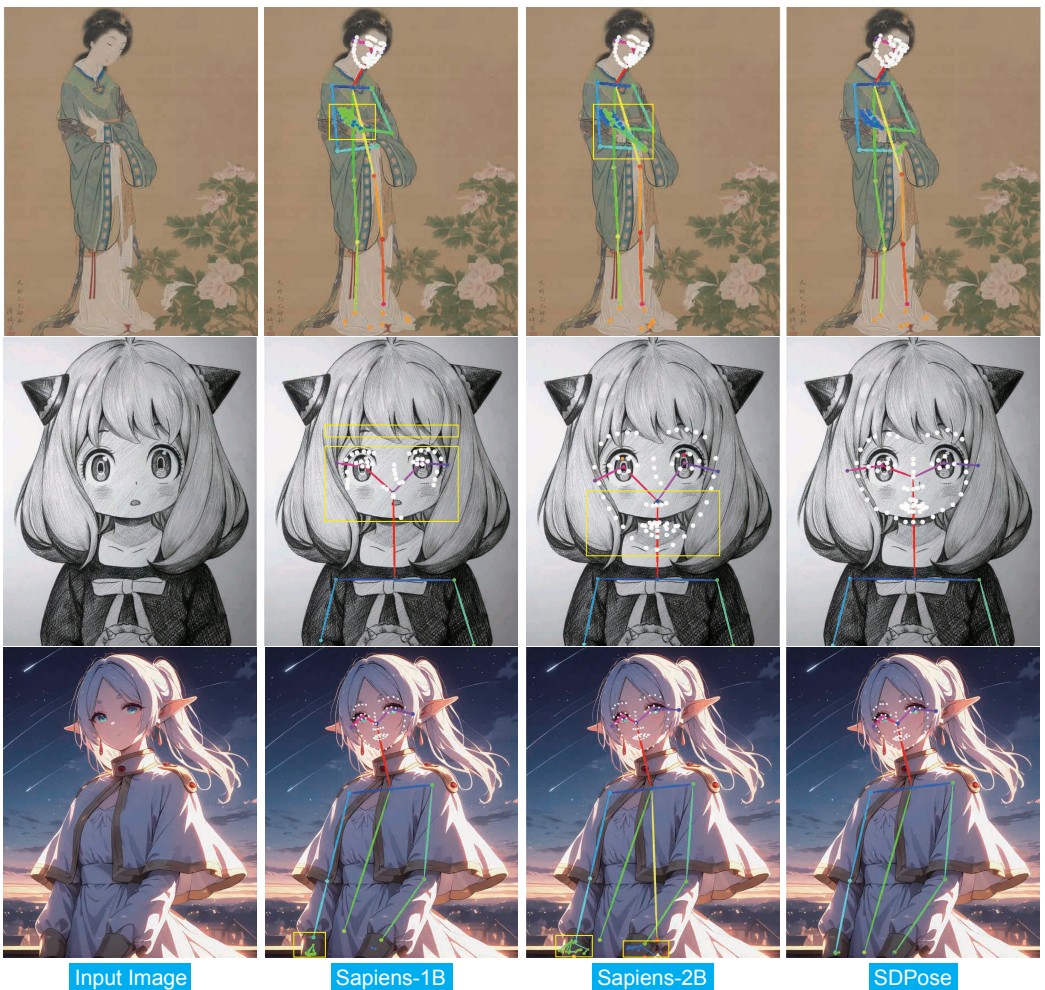

Figure 10: **Comparison on Stylized Paintings: Sapiens WholeBody vs. SDPose WholeBody.** All erroneous predictions are highlighted with yellow boxes. SDPose yields fewer false positives and notably better facial keypoint localization.

