# OpenReview forum: "$\textbf{SDPose}$: Exploiting Diffusion Priors for Out-of-Domain and Robust Pose Estimation"
_ICLR.cc/2026/Conference — Submitted to ICLR 2026_

### Official Review · Reviewer_D2Dv · 2025-10-29

**Soundness:** 2
**Presentation:** 1
**Contribution:** 2
**Rating:** 2
**Confidence:** 3

**Summary:**

SDPose is a fine-tuning framework on Stable Diffusion that repurposes pre-trained diffusion priors for human pose estimation.
While diffusion models offer rich multi-scale features and strong cross-domain robustness, their use for structured outputs like pose estimation is underexplored, and existing pose methods often degrade under domain shift and demand heavy fine-tuning.
SDPose asks whether SD U-Net latent features alone can produce reliable heatmaps.
Adopting the x₀-prediction design and the Lotus “Detail Preserver” strategy preserves fine detail and avoids overfitting, enabling efficient adaptation with minimal architectural changes.
With one-fifth of Sapiens’s training schedule, SDPose matches Sapiens-1B/2B on COCO, sets new SOTA on HumanArt and COCO-OOD.

**Strengths:**

1. By fine-tuning pre-trained diffusion models, the method achieves state-of-the-art performance in OOD pose estimation benchmark with significantly fewer training steps.

2. It also demonstrates that the intermediate features of pre-trained diffusion models are highly beneficial for generalized pose estimation.

**Weaknesses:**

1. Although prior works, as cited in the paper, have shown that the last and penultimate layers of the SD U-Net provide strong features, this study focuses on exploring features specifically for pose estimation, so conducting ablation only on these two layers seems insufficient.

2. The architecture appears to be largely borrowed from the Lotus paper, suggesting a lack of novelty.

3. For COCO-OOD generation, applying only Monet-style paintings seems limited. While Monet’s style is indeed out-of-domain, it would have been more convincing if other artistic styles were also used for style transfer.

4. The paper’s readability needs improvement.

**Questions:**

1. Much of the design seems to be inspired by the Lotus paper[1]. Could the key differences or novel contributions be clarified beyond the task difference?

2. Since it seems necessary to find an appropriate training epoch that balances pose estimation performance and generalization from the pretrained model, are there any experiments showing performance variation across training epochs?

[1] He, Jing, et al. "Lotus: Diffusion-based visual foundation model for high-quality dense prediction." arXiv preprint arXiv:2409.18124 (2024).

---

> ### Author Response · Authors · 2025-12-03
>
> ## Q1: Much of the design seems to be inspired by the Lotus paper. Could the key differences or novel contributions be clarified beyond the task difference?
> **A1:** We thank the reviewer for raising this question. Although SDPose is conceptually related to prior SD-based dense prediction work such as Lotus, our contributions extend beyond a task difference. The key distinctions are as follows:
>
> **Different problem scope and adaptation challenges.**
> Lotus focuses on continuous dense prediction (depth/normal), where supervision is pixel-aligned and geometry is directly regressed. Human pose estimation, in contrast, is a structured localization problem with discrete spatial constraints and heatmap-based supervision. Adapting diffusion U-Net latents to such structured geometric outputs requires modeling choices that differ from dense regression settings.
>
> **Multi-scale SD U-Net feature analysis and utilization.**
> Existing SD-based vision works, including Lotus, typically rely on the final latent representation or modify cross-attention layers. In SDPose, we investigate how multiple U-Net scales contribute to pose estimation and design a feature usage strategy that balances spatial precision and robustness. Our ablations show complementary behaviors across different resolutions, providing practical guidance for how diffusion latents can be adapted to structured prediction tasks.
>
> **Cross-domain evaluation enabled by COCO-OOD.**
> Pose estimation works seldom provide controlled evaluations under domain shift. We introduce COCO-OOD, a style-transferred benchmark covering several appearance variations, which allows us to systematically study generalization behavior. This setting also reveals consistent patterns in SD U-Net features, such as deeper latents being more instance-consistent and less sensitive to style variations.
>
> ---
> ## Q2: Since it seems necessary to find an appropriate training epoch that balances pose estimation performance and generalization from the pretrained model, are there any experiments showing performance variation across training epochs?
> **A2:** We thank the reviewer for the question. In our experiments, we did not observe a trade-off between the number of training epochs and OOD generalization.
>
> The training curves consistently show that validation accuracy increases rapidly during early epochs and then stabilizes. Once the model reaches this plateau, extending training for additional epochs does not degrade OOD performance within our standard training setup. In other words, we do not observe that “training longer hurts generalization”; the model’s generalization behavior remains stable across the examined training durations.

---

> ### Author Response · Authors · 2025-12-03
>
> ## W1: Although prior works, as cited in the paper, have shown that the last and penultimate layers of the SD U-Net provide strong features, this study focuses on exploring features specifically for pose estimation, so conducting ablation only on these two layers seems insufficient.
>
> **R1:** We thank the reviewer for raising this point. We would like to clarify the intended purpose of the cited works. Our intention was not to suggest that “the last two SD U-Net layers are already known to be optimal for downstream tasks.” Rather, prior studies on other large models (e.g., LLaVA-style architectures) indicate that penultimate layers often transfer better because final layers tend to encode more task-specific patterns. This motivated our initial focus on the last two SD U-Net layers, but does not imply that earlier layers are unimportant.
>
> Following the reviewer’s suggestion, the revised version now includes a complete ablation study across **all major U-Net upsampling scales (F4 → F1)**. These experiments demonstrate that **intermediate features also contribute meaningfully**, especially under domain shift.
>
> We additionally include a **block-wise t-SNE and silhouette analysis**, showing how semantic abstraction and spatial precision evolve across U-Net resolutions. This analysis helps explain why certain intermediate layers (e.g., F2) achieve a favorable balance for pose estimation.
>
> These revisions directly address the reviewer’s concern and provide a clearer understanding of how multi-scale SD U-Net features impact SDPose.
>
> ---
>
> ## W2: The architecture appears to be largely borrowed from the Lotus paper, suggesting a lack of novelty.
>
> **R2:** We thank the reviewer for the comment. As discussed in A1, our contribution does not stem from architectural changes but from:
> (i) addressing structured pose estimation under domain shift,
> (ii) analyzing **multi-scale diffusion features for structured prediction**, and
> (iii) introducing the **COCO-OOD** benchmark for controlled cross-domain evaluation.
> These aspects are not explored in Lotus or other SD-based approaches.
>
> We agree that the initial submission placed more emphasis on empirical results. In the revised manuscript, we strengthened the explanatory components by adding a new **Latent Space Analysis** section.
>
> In this analysis, we compare frozen SD-2 U-Net features with frozen Sapiens-ViT features using 300 COCO identities with three style-transferred variants each (1200 images). Silhouette scores and t-SNE visualizations reveal:
>
> 1. **Sapiens features cluster primarily by style**, indicating stronger sensitivity to appearance variations.
> 2. **Diffusion U-Net features cluster primarily by person-instance**, showing more geometry-centric and style-invariant representations.
>
> This contrast offers insight into why diffusion backbones maintain robustness across artistic and domain shifts. The full analysis is included in Section 5.4 of the revised version.
>
> ---
>
> ## W3: For COCO-OOD generation, applying only Monet-style paintings seems limited. Using additional artistic styles would be more convincing.
>
> **R3:** We appreciate the reviewer’s helpful suggestion. We agree that relying on a single style is limiting, and we have extended COCO-OOD accordingly.
>
> The revised benchmark now includes three stylistically diverse variants:
>
> - **Ukiyo-e**, generated using CycleGAN-based transformations;
> - **Color Sketch**, generated using Nano-Banana;
> - **Monet/Oil**, the original StyTr2-based variant.
>
> These additions create a **multi-style, multi-model OOD benchmark** that reduces single-style bias and enables a more comprehensive evaluation of cross-domain robustness.
>
> ---
>
> ## W4: The paper’s readability needs improvement.
>
> **R4:** We thank the reviewer for this feedback. In the revised version, we improved readability by:
>
> - restructuring the Introduction as well as Sections 4 and 5 to provide a clearer logical flow and smoother transitions,
> - refining key paragraphs for clarity and consistency,
> - unifying table formats and improving figure layouts for easier comparison.
>
> These changes enhance the overall accessibility of the paper.

---

### Official Review · Reviewer_PwGQ · 2025-10-30

**Soundness:** 3
**Presentation:** 3
**Contribution:** 2
**Rating:** 2
**Confidence:** 5

**Summary:**

This paper introduces SDPose, a novel framework that adapts a pre-trained Stable Diffusion (SD) model for human pose estimation. The core idea is to leverage the rich, general-purpose visual features learned by the SD model's U-Net to achieve superior robustness, especially on out-of-domain (OOD) data like artistic images. Instead of modifying the cross-attention mechanisms or adding new embeddings, SDPose makes minimal changes: it adds a lightweight convolutional head to predict keypoint heatmaps directly from the U-Net's intermediate features and uses an auxiliary RGB reconstruction task to prevent overfitting and preserve the model's generative priors.

**Strengths:**

- It targets the OOD task and finds a simple method to use the pretrained model for the OOD task. It  shows the application of SDPose as a zero-shot pose annotator for ControlNet image generation and video generation provides tangible evidence of its superior qualitative performance over baselines like DWPose
- The paper is well organized and evaluations are comprehensive.

**Weaknesses:**

- My major concern is the novelty of this work. The novelty of this paper is quite marginal. It just uses the pretrained model for a pose estimation task. Nothing seems special or challenging.
- This paper just did the experiments but not provide some insights. It would be better to provide some explanations about why the proposed method works better.

**Questions:**

Refer to strengths and weaknesses

---

> ### Author Response · Authors · 2025-12-03
>
> # Reviewer 3: PwGQ
>
> We sincerely thank the reviewer for their thoughtful and constructive feedback, which has greatly helped us refine the quality and clarity of our work. We address the reviewer’s concerns as follows:
>
> ## Q/W1: My major concern is the novelty of this work. The novelty of this paper is quite marginal. It just uses the pretrained model for a pose estimation task. Nothing seems special or challenging.
>
> We respectfully disagree with the reviewer’s assessment. Although SDPose builds upon a pretrained diffusion model, as many recent works in dense prediction do, the novelty of our work does not lie in modifying the diffusion architecture, but rather in **exploring aspects of diffusion priors that have been largely unexplored in prior work for pose estimation**. Our contributions are three-fold:
>
> ### 1. **Cross-domain pose estimation is an underexplored but practically important problem.**
> Most existing pose estimators are trained and evaluated within a single domain (COCO → COCO). However, real-world settings frequently involve domain shifts such as artistic renderings, stylized images, anime. Very few works study how pose models behave under these shifts.
>
> Our work shows that diffusion U-Net representations encode rich pose-related geometry that transfers significantly better than ViT/CNN backbones. Addressing this problem setup is both underexplored and non-trivial.
>
> ### 2. **We provide a systematic analysis of multi-scale SD U-Net latent features for structured prediction.**
> Prior diffusion-based adaptations mainly fall into two categories:
> (1) using only the **final** U-Net latent features, or
> (2) modifying **cross-attention layers**, an intrusive architectural change.
>
> To our knowledge, these works do not directly examine how intermediate SD U-Net features benefit downstream structured prediction tasks.
> SDPose performs a **comprehensive F4→F1 analysis**, revealing distinct trade-offs between semantic robustness and spatial precision. This multi-scale analysis does not exist in Marigold, Lotus, GenLoc, or related work.
>
> ### 3. **We introduce a systematic OOD benchmark (COCO-OOD) and representation analysis for pose.**
> Existing pose estimation works rarely assess cross-domain robustness.
> To fill this gap and complement existing benchmarks like HumanArt, we construct **COCO-OOD**, which spans multiple style-shift domains (Ukiyo-e, Color Sketch, Monet-Oil).
> This reveals a new phenomenon: SD features cluster primarily by person instance, whereas Sapiens ViT features cluster primarily by style, highlighting an intrinsic robustness property of diffusion priors that, to our knowledge, has not been characterized before.
>
> ### **Summary**
> While SDPose leverages a pretrained diffusion model as its backbone, the novelty lies in:
> (i) addressing the **underexplored but important** problem of cross-domain pose generalization,
> (ii) delivering the **comprehensive analysis of multi-scale SD U-Net features** for structured prediction, and
> (iii) introducing a **systematic OOD benchmark + representation analysis** explaining why diffusion priors outperform ViT/CNN backbones.
>
> ## Q/W2: This paper just did the experiments but not provide some insights. It would be better to provide some explanations about why the proposed method works better.
>
> **A2:** We thank the reviewer for highlighting this important point. In the revised version, we added a dedicated **latent-space analysis** section (Section 5.4) to offer deeper insight into why SDPose exhibits strong cross-domain robustness.
>
> Specifically, we compare features extracted from the pretrained SD-2 U-Net with those from the pretrained Sapiens backbone, with neither model fine-tuned on pose labels. Using 300 COCO instances, each rendered into three additional style-transferred variants (1200 images total), we compute silhouette scores and visualize the feature distributions using t-SNE. The results reveal the following key insights:
>
> 1. **SD U-Net features cluster primarily by person instance rather than image style**, indicating strong invariance to appearance and texture shifts.
> 2. **Sapiens features cluster mainly by style**, suggesting that appearance cues dominate its representation and weaken cross-domain generalization.
> 3. **SD multi-scale features retain geometric consistency across domains**, helping SDPose maintain stable keypoint localization even under significant style changes.
>
> These analyses provide both intuitive and quantitative explanations for why diffusion-based representations lead to improved robustness, complementing the empirical findings presented in the main paper.

---

### Official Review · Reviewer_7EQQ · 2025-10-31

**Soundness:** 4
**Presentation:** 4
**Contribution:** 4
**Rating:** 8
**Confidence:** 4

**Summary:**

This paper presents SDPose, a fine-tuning framework that leverages Stable Diffusion (SD) as a vision backbone for human pose estimation. Instead of modifying cross-attention layers or using learned embeddings, SDPose directly predicts keypoint heatmaps in the SD U-Net’s latent space, preserving the generative priors of diffusion models. The authors also construct COCO-OOD, a style-transferred variant of COCO for systematic robustness evaluation. SDPose matches or surpasses Sapiens-1B/2B on COCO with only one-fifth of the training cost and achieves new state-of-the-art performance on HumanArt and COCO-OOD. The model further demonstrates zero-shot pose annotation capabilities for pose-guided image and video generation, establishing SDPose as an efficient, generalizable, and versatile framework for structured prediction using diffusion priors

**Strengths:**

- Novel approach: Exploits Stable Diffusion’s latent features directly, preserving generative priors instead of introducing ad-hoc conditioning modules.
- Strong performance: Matches SoTA on COCO and achieves new records on HumanArt and COCO-OOD, with significant training efficiency (1/5 of Sapiens training time)
- Benchmark contribution: Introduces COCO-OOD, enabling standardized OOD robustness evaluation for pose models.

**Weaknesses:**

- Analysis of computational cost: The paper highlights training efficiency but provides limited discussion of inference-time latency and memory overhead compared to conventional backbones like ViTPose.
- Ablation breadth: Some architectural design choices (e.g., selection of diffusion timestep or use of SD-v1.5 vs v2) could benefit from more justification or quantitative analysis.

**Questions:**

- How would SDPose perform under multi-person scenarios or crowded scenes, where top-down cropping might limit context?
- Would fine-tuning from more recent foundation diffusion models (e.g., SDXL) further improve generalization, or is the performance already saturated?

---

> ### Author Response · Authors · 2025-12-03
>
> # Reviewer 2 7EQQ:
>
> We sincerely thank the reviewer for their positive and encouraging feedback on our work.
>
> ## Q1: How would SDPose perform under multi-person scenarios or crowded scenes, where top-down cropping might limit context?
>
> **A1:** Thank you for raising this important question. We address it from three perspectives:
>
> 1. **COCO already includes many crowded scenes.**
>    Although SDPose adopts a top-down pipeline, COCO contains a substantial number of multi-person and highly crowded instances. SDPose is trained and evaluated on this distribution, and its performance already reflects robustness to overlapping persons.
>
> 2. **Crowded-scene limitations are dominated by the detector, not the pose head.**
>    In very dense scenarios, the accuracy of any top-down approach is primarily constrained by the bounding-box detector rather than the pose estimator. Improving detector robustness in high-density settings is a valuable direction but outside the scope of this work. We appreciate the reviewer’s suggestion and are interested in extending SD-based representations to detection in future work.
>
> 3. **Top-down methods typically retain sufficient context for pose estimation.**
>    In top-down pipelines, the detector-generated crop naturally preserves enough local context around each person for accurate keypoint prediction. This design is widely adopted in state-of-the-art pose estimators and has been shown to work reliably even in multi-person scenes. SDPose follows the same principle, and we do not observe qualitative failures attributable to missing context.
>
>
> ## Q2: Would fine-tuning from more recent foundation diffusion models (e.g., SDXL) further improve generalization, or is the performance already saturated?
>
> **A2:** We appreciate the reviewer’s insightful question. Extending SDPose to more recent diffusion foundations such as SDXL is indeed a promising direction. We did not include SDXL-based experiments in this submission mainly due to computational constraints, as SDXL requires significantly heavier resources (e.g., a two-UNet architecture and a much larger text encoder).
> Conceptually, we expect SDXL’s stronger priors, with higher-capacity UNet, richer latent space, and higher-quality training distribution, to further enhance cross-domain robustness. Although this benefit has not been empirically validated in our pipeline, similar trends have been observed in prior works. For example, the performance gap between Lotus [1] and Lotus-2 [2] partially stems from upgrading the underlying generative backbone, suggesting that stronger diffusion foundations can provide more expressive priors for downstream tasks.
> Nevertheless, we avoid making claims without direct evaluation. We view SDXL integration as valuable future work and plan to investigate whether its stronger generative priors yield measurable gains for robustness-oriented pose estimation.
>
> ## W1: Analysis of computational cost: The paper highlights training efficiency but provides limited discussion of inference-time latency and memory overhead compared to conventional backbones like ViTPose.
>
> **R1:** Our comparison primarily focuses on parameter count, which directly reflects the computational scale of the VAE and U-Net backbone dominating inference. There is an inherent trade-off: recent SOTA models such as Sapiens (2B params) rely on large backbones for generalization, whereas SDPose achieves comparable robustness with a significantly smaller (0.9B) model.
>
> ## W2: Ablation breadth: Some architectural design choices (e.g., selection of diffusion timestep or use of SD-v1.5 vs v2) could benefit from more justification or quantitative analysis.
>
> **R2:** We thank the reviewers for pointing this out. Regarding the timestep choice, prior diffusion-based dense prediction work [1] has already extensively studied the effect of different noise levels and timestep schedules (e.g., single-step vs. multi-step, early vs. late timesteps). These findings consistently show that timestep choices yield relatively minor variations compared to architectural factors.
>
> Given this, we intentionally shift our focus to how multi-scale diffusion features contribute to pose representation, which is a direction not explored in prior work. Our goal is to understand which feature encode style-invariant info and how these features can be effectively aggregated, which is an aspect that has been largely overlooked in previous diffusion-based adaptations.
>
> [1] Lotus: Diffusion-based visual foundation model for high-quality dense prediction. J He, H Li, W Yin, Y Liang, L Li, K Zhou, H Zhang, B Liu, YC Chen. ICLR, 2025.
>
> [2] Lotus-2: Advancing Geometric Dense Prediction with Powerful Image Generative Model. J. He, H. Li, M. Sheng, Y.-C. Chen. arXiv preprint arXiv:2512.01030, 2025.

---

### Official Review · Reviewer_TtNN · 2025-11-01

**Soundness:** 3
**Presentation:** 3
**Contribution:** 3
**Rating:** 4
**Confidence:** 3

**Summary:**

This paper proposes SDPose, a fine-tuning framework built upon Stable Diffusion (SD) to adapt pre-trained diffusion priors for 2D human pose estimation, particularly focusing on out-of-domain robustness. SDPose operates entirely in the latent space of the SD U-Net without modifying attention modules. The authors add a lightweight heatmap head to map multi-scale latent features to keypoint heatmaps and an auxiliary RGB reconstruction branch for generative regularization, preserving domain-transferable visual semantics. Additionally, the authors introduce COCO-OOD, a style-transferred version of COCO (via CycleGAN), to benchmark robustness under domain shift. Experiments show SDPose achieves SOTA performance on COCO, HumanArt, and COCO-OOD. It also demonstrates zero-shot usability as a pose annotator for controllable image/video generation.

**Strengths:**

1. The paper, unlike prior adaptations (e.g., fine-tuned cross-attention or learned condition embeddings), leverages pre-trained diffusion features with minimal architectural change, preserving SD’s representational power. The auxiliary RGB reconstruction task is a simple yet effective regularization strategy for maintaining generative semantics during fine-tuning.
2. The authors show SD Pose almost matches Sapiens-1B/2B on COCO while requiring 1/5 training epochs and a smaller backbone compared to the 2B backbone. The paper also shows SOTA performance on other in-the-wild datasets - HumanArt and COCO-OOD. Ablations validating the contribution of diffusion priors and auxiliary reconstruction are helpful.
3. Experimental tables are well-organized and reproducibility details (hardware, hyperparameters, dataset splits) are appreciable.

**Weaknesses:**

1. The core idea of repurposing SD U-Net features for downstream vision is shared with prior works like Marigold, Lotus, and GenLoc. The proposed approach’s novelty lies mainly in its architectural restraint (no attention tuning) and addition of a reconstruction branch.
2. The paper does not deeply analyze why diffusion priors confer robustness, e.g., what specific latent feature properties (multi-scale, texture invariance, semantic richness) contribute most.
3. OOD tests (HumanArt, COCO-OOD) are style-based and geometric or camera-domain shifts (e.g., occlusion, lighting, or viewpoint) are not considered.

**Questions:**

1. Would SDPose generalize to other OOD types such as blur, occlusion or synthetic-to-real transfer? Could the authors test or discuss applicability in non-artistic domain shifts?
2. Did the authors explore multi-scale fusion from different U-Net layers, rather than selecting a single feature map?

---

> ### Author Response · Authors · 2025-12-03
>
> # Reviewer 1 TtNN:
>
> We sincerely thank the reviewers for their positive, encouraging, and constructive feedback. We appreciate the recognition that SDPose preserves the representational strengths of Stable Diffusion with minimal architectural modification, achieves strong efficiency gains (e.g., 1/5 of Sapiens’ training epochs), and delivers state-of-the-art performance on cross-domain pose estimation benchmarks. We also thank the reviewers for highlighting the soundness of our experiments and the completeness of our reproducibility details.
>
> ## Q1: Would SDPose generalize to other OOD types such as blur, occlusion or synthetic-to-real transfer? Could the authors test or discuss applicability in non-artistic domain shifts?
>
> **A1**: We thank the reviewer for raising this important point. Our work focuses on artistic and style-induced domain shifts, where diffusion priors provide the most distinctive advantage over conventional backbones. For other non-artistic OOD factors such as blur, occlusion, or mild photometric distortions, modern top-down pose estimators (e.g., Sapiens Pose [1]) typically rely on data-augmentation pipelines to achieve robustness. SDPose follows the same augmentation protocol during fine-tuning, and therefore inherits the same resilience to such perturbations.
>
> ## Q2: Did the authors explore multi-scale fusion from different U-Net layers, rather than selecting a single feature map?
>
> **A2**: We appreciate the reviewer’s suggestion. In our revised submission, we have expanded the ablation to cover **all four upsampling layers (F4 → F1)** of the SD U-Net (see Table 2), which directly addresses the concern about only examining a single feature level. This analysis shows a clear trade-off across scales: deeper blocks (F3/F4) encode stronger style-invariant and instance-consistent semantics but operate at coarse resolutions, while F2 offers the best balance of semantic robustness and spatial precision under domain shift.
>
> Regarding multi-scale fusion, our design intentionally focuses on isolating the contribution of **pretrained diffusion features**. Fusion modules such as FPN-style necks introduce large numbers of randomly initialized parameters that are not part of SD’s generative pretraining, making it difficult to attribute performance gains to the diffusion prior itself. To maintain a clean and interpretable analysis, we therefore use a single feature level.
>
> We agree that multi-scale fusion is a valuable architectural extension and plan to explore it in future work, especially after establishing the single-scale behaviors of SD features in this paper.
>
> [1] Sapiens GitHub configuration: https://github.com/facebookresearch/sapiens/blob/main/pose/configs/sapiens_pose/coco/sapiens_1b-210e_coco-1024x768.py#L135

---

> > ### Author Response · Authors · 2025-12-03
> >
> > ## W1: The core idea of repurposing SD U-Net features for downstream vision is shared with prior works like Marigold, Lotus, and GenLoc. The proposed approach’s novelty lies mainly in its architectural restraint (no attention tuning) and addition of a reconstruction branch.
> >
> > We appreciate the reviewer’s comment. While we share the general motivation of leveraging diffusion priors, the prior works mentioned—Marigold, Lotus, and GenLoc—operate in fundamentally different problem settings and do not explore the aspects that SDPose focuses on.
> >
> > **(1) Task-level difference.**
> > Marigold and Lotus address dense prediction tasks such as depth or normal estimation, where the output is a continuous pixel-aligned regression field. These works rely primarily on the *final* SD U-Net features produced by the last convolutional block and do not involve structured geometric constraints. GenLoc, although related to keypoint reasoning, performs object-level localization using *cross-attention maps and learned conditional embeddings*, and requires modifying the attention layers.
> >
> > **(2) Limited prior exploration of multi-scale SD U-Net representations.**
> >
> > Existing diffusion-prior works almost exclusively use the final U-Net output or cross-attention maps.
> > To the best of our knowledge, prior work has not systematically examined how intermediate, multi-scale SD U-Net latent features behave on downstream vision tasks, nor how different layers trade off semantic robustness and spatial precision for structured prediction.
> > SDPose provides the first comprehensive (F4→F1) multi-scale ablation in this context, revealing how different SD layers contribute differently to pose estimation—an analysis that, to our knowledge, is absent in existing literature.
> >
> > **(3) Lack of prior explanation of diffusion priors’ robustness.**
> > Previous works largely report empirical robustness without analyzing *why* diffusion features generalize well across domains. In contrast, we provide a **latent-space analysis** (t-SNE + silhouette scores) comparing SD U-Net features with a strong ViT backbone, showing that SD latents form *instance-consistent and style-invariant clusters*. This offers a concrete explanation for the cross-domain robustness of diffusion priors, which has not been explored before.
> >
> > **Summary.**
> > Our contribution therefore goes beyond architectural restraint. SDPose fills an unexplored gap by:
> > (i) systematically studying how diffusion priors transfer to **structured pose estimation**,
> > (ii) performing the comprehensive analysis of **multi-scale U-Net latent features** for downstream pose estimation, and
> > (iii) providing the **analytical explanation** for diffusion priors’ cross-domain robustness.
> > These aspects have not been addressed in Marigold, Lotus, or GenLoc.
> >
> > ## W2: The paper does not deeply analyze why diffusion priors confer robustness, e.g., what specific latent feature properties (multi-scale, texture invariance, semantic richness) contribute most.
> >
> > **R2:** We thank the reviewer for raising this point. In the revised version, we added a dedicated **latent-space analysis** (Section 5.4) section to provide deeper insight into why diffusion priors exhibit strong robustness under domain shifts.
> >
> > We compare features extracted from the pretrained SD-2 U-Net with those from the pretrained Sapiens backbone. Using 300 COCO identities with four style variants each (1200 images total), we compute silhouette scores and visualize the feature distributions with t-SNE.
> >
> > The results show a clear and consistent pattern:
> >
> > - **Sapiens ViT features cluster primarily by style**, indicating high sensitivity to appearance and texture changes.
> > - **SD U-Net features cluster primarily by person instance**, demonstrating that diffusion latents encode **style-invariant structure** across multiple scales.
> >
> > Together, these observations provide direct evidence that the cross-domain robustness of diffusion priors arises from their ability to preserve instance-level structure while suppressing stylistic factors.
> >
> > We have added these quantitative metrics, visualizations, and interpretations to the revised manuscript (Section 5.4), addressing the reviewer’s concern regarding the source of robustness.
> >
> > ## W3: OOD tests (HumanArt, COCO-OOD) are style-based and geometric or camera-domain shifts (e.g., occlusion, lighting, or viewpoint) are not considered.
> >
> > **R3**: As discussed in A1, modern pose estimators already incorporate robustness to geometric and photometric perturbations such as occlusion, blur, and lighting. SDPose builds on the same pipeline and therefore inherits these robustness properties. Our work focuses specifically on style-based OOD settings, which are not covered by existing baselines.

---

### Author Response · Authors · 2025-12-03
**Summary for Area Chair**

We sincerely appreciate the reviewers’ thoughtful and constructive feedback. Their comments were extremely valuable in helping us refine the presentation and sharpen the framing of our contributions. Below, we summarize the key points we hope the AC will take away:

1. **Positive Feedback**
   We thank the reviewers for their constructive and encouraging feedback. Across all reviews, there is a clear consensus that our experimental evaluation is **sound, solid, and comprehensive**, and that our work provides **meaningful engineering contributions** supported by extensive empirical evidence. We appreciate that the reviewers recognized the **practical significance** of our approach and its **potential impact on cross-domain pose estimation**.

2. **Clarifications on Our Contributions and Novelty**
   Our contributions are **fundamentally distinct** from previous works such as Marigold and Lotus, which adapt diffusion priors primarily for **dense prediction tasks** like depth or normal estimation. In contrast:
   - **Task novelty:** Using diffusion priors for pose estimation is underexplored and technically distinct from pixel-level dense prediction.
   - **Multiscale SD-U-Net feature analysis:** We provide the first systematic study on how different SD U-Net latent levels affect pose estimation, including t-SNE evidence showing SD priors yield more style-invariant and identity-consistent features than Sapiens ViT.
   - **COCO-OOD benchmark construction:** We construct **COCO-OOD**, a style-transferred extension of COCO that enables systematic evaluation under domain shifts.

   Together, these innovations form a coherent framework that not only offers a new perspective on leveraging diffusion priors for structured prediction, but also directly translates into **state-of-the-art cross-domain robustness** on pose estimation benchmarks.

3. **Reviewers’ Core Concerns**

  The reviewers raised several central questions and requests for clarification:

- **Novelty and Contribution.**
  The reviewer expressed concerns that SDPose may appear architecturally similar to works such as Lotus and Marigold, and that the main distinction may be the task domain.

- **Scope of OOD Evaluation.**
  The reviewer noted that our current OOD evaluation includes only a single artistic style-transfer variant and suggested incorporating additional styles.

- **Explanation of Robustness.**
  The reviewer requested deeper theoretical or analytical insight into *why* diffusion priors exhibit robustness under domain shift, beyond empirical performance alone.

- **Multi-scale Feature Analysis.**
  The reviewer noted that the ablation analysis was limited to the last two SD U-Net feature levels and suggested extending it to include all upsampling blocks for a more complete evaluation.

4. **Supplemented Experiments and Additional Improvements**
   Compared to the first version of the submission, we have added the following extensions to directly address the reviewers’ concerns:

   (i) **Enhanced COCO-OOD Benchmark.**
   We expanded the COCO-OOD benchmark by incorporating *Ukiyo-e* and *Color Sketch* style variants generated by multiple style-transfer models, in addition to the original Monet-Oil version. This provides a more comprehensive and diverse evaluation of cross-domain robustness beyond a single stylistic shift.

   (ii) **Latent-Space Analysis Explaining Diffusion Priors’ Robustness.**
   To understand why diffusion priors are robust under domain shifts, we conducted a detailed t-SNE analysis comparing **multi-scale SD U-Net latent features** and **Sapiens ViT features** across real and style-transferred domains.
   We observe that SD U-Net features form **identity-consistent and style-invariant clusters**, whereas Sapiens ViT features are **much more sensitive to stylistic perturbations**.
   This analysis provides an interpretable explanation for *why diffusion priors exhibit stronger OOD generalization* in pose estimation.

   (iii) **Comprehensive Ablation Study on SD U-Net Multi-Scale Features.**
    We added a detailed ablation over the four upsampling features (F4 to F1) from the SD U-Net. Together with our latent-space analysis—showing that deeper blocks encode stronger identity-aware but much coarser, low-resolution features—this ablation highlights a key trade-off between semantic robustness and spatial precision. This analysis justifies selecting F2 as the default feature level for SDPose under cross-domain settings, while the last block remains optimal for purely in-domain scenarios.

   (iv) **Improved Readability and Presentation.**
   We revised the writing, reorganized several sections, and updated the formatting of multiple tables to provide a clearer and more reader-friendly presentation. These adjustments make the results easier to interpret and improve the overall accessibility of the paper.

---

### Meta-Review · Area_Chair_hUuH · 2026-01-07

**Summary:**

The paper proposes SDPose, a diffusion-based framework for robust human pose estimation under domain shift, leveraging Stable Diffusion U-Net features and an auxiliary RGB reconstruction branch, and introduces the COCO-OOD benchmark.

Reviewers generally agree that the paper is well written, experimentally thorough, and demonstrates strong empirical performance, particularly on style-based out-of-domain benchmarks.

However, the overall consensus is that the conceptual novelty over prior diffusion-prior works (e.g., Marigold, Lotus, GenLoc) is limited, with the main differentiation being task adaptation rather than a fundamentally new methodology.

While the rebuttal added analyses and experiments, these improvements were viewed as incremental clarifications rather than decisive responses to the core concerns. As a result, the paper does not clearly meet the acceptance bar.

**Reviewer Concerns:**

Concerns addressed by the rebuttal:
- Multi-scale SD U-Net feature usage: Extended ablations across multiple upsampling blocks clarified design choices and trade-offs.
- Explanation of robustness: Added t-SNE and silhouette analyses provided a more concrete empirical explanation of why diffusion priors yield style-invariant features.
- OOD evaluation breadth (style-based): COCO-OOD was expanded to include multiple artistic styles, strengthening the evaluation within this setting.
- Presentation and clarity: Writing, tables, and experimental details were improved.

Outstanding concerns:
- Novelty: The approach is still perceived as an incremental extension of existing diffusion-prior adaptations, with limited conceptual departure beyond task application.
- Scope of robustness: OOD evaluation remains largely restricted to artistic/style shifts; robustness to geometric, camera, or physical perturbations is not convincingly demonstrated.
- Methodological depth: The contribution is seen as primarily engineering-driven, without a sufficiently strong new modeling insight to justify acceptance at this venue.

**Reviewer Scores:**

- Reviewer TtNN: Likely remains marginal reject (4) , appreciates experiments and clarifications but maintains concerns on novelty and scope.
- Reviewer 7EQQ: Likely remains accept poster (8), very positive on performance, efficiency, and benchmark contribution.
- Reviewer D2Dv and PwGQ: Likely remains reject,  acknowledges strengths but unconvinced that rebuttal resolves core novelty concerns.

Overall: Minor upward shifts in confidence and clarity after rebuttal, but insufficient to overturn the mixed-to-negative consensus, leading to a rejection decision.

---

### Decision · Program_Chairs · 2026-01-26

Reject